# Genomic Analysis of the Natural Variation of Fatty Acid Composition in Seed Oils of *Camelina sativa*

**DOI:** 10.3390/biology14091199

**Published:** 2025-09-05

**Authors:** Samuel Decker, Wilson Craine, Timothy Paulitz, Chengci Chen, Chaofu Lu

**Affiliations:** 1Department of Plant Sciences and Plant Pathology, Montana State University, Bozeman, MT 59717, USA; samuel.decker@student.montana.edu; 2Department of Crop and Soil Sciences, Washington State University, Pullman, WA 99164, USA; wilson.craine@wsu.edu; 3Wheat Health Genetics and Quality Research Unit, US Department of Agriculture—Agricultural Research Service, Pullman, WA 99164, USA; 4Eastern Agricultural Research Center, Montana State University, Sidney, MT 59270, USA; cchen@montana.edu

**Keywords:** *Camelina sativa*, genetic marker, GWAS, fatty acids, oilseed, pangenome, population structure, QTL

## Abstract

Camelina is an emerging oilseed plant mainly used for biofuel production. Fatty acid composition is an important trait that affects the oil properties of plants. Although much is known about the mechanisms of fatty acid metabolism in plants, a knowledge gap remains regarding the unique characteristics of camelina oil. With modern genome-wide association studies using advanced genomic tools and natural variation, one can uncover genetic mechanisms that determine complex traits of interest in an organism. This approach has been applied to camelina in traits like seed oils, but with limited success due to the quality of a reference genome and the small range of variation in fatty acids in natural populations. This study aimed to uncover novel genetic mechanisms of fatty acid metabolism in camelina seed. The ability to achieve this goal was enhanced by developing abundant molecular markers using an improved camelina genome and capturing the greater variation of fatty acids when plants were grown under different environments, such as geographical locations and nitrogen fertilization regimes. This study resulted in 118 markers associated with fatty acid contents across environments, providing a launchpad to discover potentially novel genes that affect the quality of camelina oils.

## 1. Introduction

Camelina (*Camelina sativa* L. Crantz) is an oilseed crop that has gained popularity in recent years primarily due to its potential to provide a sustainable feedstock for biofuels and as an intermediate cover crop (Taheripour et al.) [1]. Extensive research has been conducted to improve camelina characteristics [2]. A major goal of camelina breeding is to optimize the seed oil fatty acid composition and meet the requirements of different end uses. Camelina seed typically contains 30–40% oil (triacylglycerol, or TAG) and 25–30% protein by weight, depending on genotype and environmental conditions during seed filling [3]. Camelina oil contains 50–60% polyunsaturated fatty acids (PUFAs), including linoleic (18:2) and omega-3 linolenic (18:3) acids [4] that are essential fatty acids for mammals, thus making the oil an excellent nutritional source for humans and livestock. However, these PUFAs are not desirable for industrial uses such as biofuels due to their oxidative instability. Studies indicate that increasing the proportion of monounsaturated oleic acid (*cis*-9-octadecenoic acid, 18:1) in vegetable oils provides significantly improved oxidative stability [5]. Hence, high-oleic, low-PUFA oils are a general target of oilseed breeding for industrial purposes [6]. Conventional breeding and modern biotechnological approaches have been applied to modify the fatty acid composition in camelina oils. Our work on mutagenesis using the chemical EMS (Ethyl Methane Sulfonate) resulted in lower PUFAs (45% vs. 54% in WT) and correspondingly increased oleic acid (from 17% to 27%) due to the mutation at one *FAD2* locus in the hexaploid genome [4]. 54% in WT) and correspondingly increased oleic acid (from 17% to 27%) due to the mutation at one *FAD2* locus in the hexaploid genome [4]. Another mutation at an *FAE1* locus caused an over 60% reduction of the very-long chain fatty acids (VLCFAs), primarily eicosenoic acid (20:1) and erucic acid (22:1), compared to the wildtype [7]. Biotechnological approaches have also been applied in modifying camelina oil profiles. CRISPR knockout of all three *FAE1* alleles nearly eliminated (<1%) the VLCFAs and increased the 18:3 level from 39% to 50% [7]. Both RNAi [4] and CRISPR [8] targeting the *FAD2* genes increased oleic acid (from 16% to over 50%) and decreased PUFAs significantly (from 50% to 16%). However, knocking out all *FAD2* alleles severely inhibited plant growth [9]. Achieving a satisfying fatty acid composition without compromises in plant growth and crop productivity remains a challenge. Therefore, it is essential to gain a comprehensive understanding of the mechanisms of fatty acid biosynthesis and oil accumulation in camelina seed.

Seed lipid metabolism in camelina shares many mechanisms described in *Arabidopsis thaliana* and oilseed crops [10]. However, some aspects are unique in camelina as several genetic regulators were identified that are involved in the control of fatty acid metabolism and TAG accumulation [11]. Camelina oil has a uniquely high linolenic acid (18:3) content. A dramatic surge of 18:3 accumulation was observed during the later stages of seed development. Once 18:3 reached a peak, there was an abrupt cessation of accumulation, and the 18:3 level slightly declined; however, the bulk of the accumulated 18:3 remained during the rest of seed maturation [12]. The VLCFA 20:1 accumulation traced that of 18:3, though at a much lower level, while the accumulation of 18:1, the common precursor of desaturation and elongation, concurrently declined and remained relatively low [12]. Besides these findings, biotechnological experiments also suggested unique features of fatty acid metabolism in camelina seed. Metabolic engineering of fish oil-type long-chain PUFAs, including eicosapentaenoic acid (EPA) and docosahexaenoic acid (DHA), was achieved in Arabidopsis by reconstructing the biosynthetic pathway comprising heterologous fatty acid desaturase and elongase genes [13]. The high levels of such LC-PUFAs that matched those found in marine fish oils were only achieved by expressing those genes in camelina, suggesting the contribution of endogenous seed lipid mechanisms in camelina for efficient accumulation of DHA and EPA [14]. The LC-PUFA levels in camelina seed were further enhanced by eliminating the FAE1-mediated elongation pathway [15]. In another recent experiment, overexpression of a codon-optimized FAD2 and the *Brassica napus* FAD3 resulted in an ultra-high accumulation of 18:3 (over 68%) in the camelina *fae1* mutant background (39%), compared to a lower level at 57% in the Arabidopsis *fae1* background transformed with the same construct [16].

Previous studies of a collection of the camelina germplasm in field trials revealed the quantitative variation in seed oil and fatty acid contents. This allowed for a genome-wide association study (GWAS) to reveal more genetic mechanisms of camelina seed lipid metabolism. However, limited quantitative trait loci (QTLs) were identified for seed oil traits, including only one fatty acid, the 18:1 content [17]. Besides the limitation due to the low diversity of the germplasm panel [18], we hypothesized that more QTLs can be discovered by increasing the marker density, and, more importantly, capturing greater phenotypic variation by growing plants in multiple environments. In this study, we developed high-density molecular markers using a recently sequenced camelina reference genome of Suneson, which was much improved over the one developed over a decade ago [19] that was used in our previous study [17]. We also grew the population in multiple locations under different regimes of nitrogen fertilization. The results not only validated our previous findings but also provided new QTLs for several fatty acids besides 18:1. In addition, the study revealed interesting correlations among seed oil traits and different environmental effects on the accumulation of different fatty acids in camelina seed. These results provide valuable resources to discover novel mechanisms of fatty acid metabolism. They may also lead to novel strategies to improve camelina seed oil qualities and develop superior varieties adapted to various growth environments for desirable fatty acid profiles.

## 2. Materials and Methods

### 2.1. Plant Materials and Field Experiments

The diversity panel used in this study, comprising 212 *Camelina sativa* accessions collected worldwide, including wild and cultivated varieties and breeding lines, was described in the previous report [17]. The population was grown in four geographical locations with diverse climate conditions, including Bozeman, MT, USA (Latitude 45.67, Longitude −111.15), Sidney, MT (47.43, −104.09), Pullman, WA (46.75, −117.10), and Wuhan, China (30.47, 114.35). Annual precipitation from rain and snow for each location during the 2022 growing season was 81.8 cm in Bozeman, 43.7 cm in Sidney, 79.8 cm in Pullman, and 113.3 cm in Wuhan, respectively. The average high and low temperatures during the growing season (May–September) were 23.6–6.9 °C in Bozeman, 25.9–10.9 °C in Sidney, 24.4–8.8 °C in Pullman, and 29.1–23.1 °C in Wuhan. The average daily incident shortwave energy for each location during the growing season was 6.66 kWh in Bozeman, 6.34 kWh in Sidney, 6.44 kWh in Pullman, and 5.44 kWh in Wuhan. Additional environmental information at each location is contained in Appendix A. Besides field trials in Bozeman and Wuhan in 2018, plants were also grown in Sidney and Pullman in 2022–2023. In new sites at Sidney and Pullman fields, additional nitrogen (N) fertilizers were applied at 100 kg N ha^−1^ (high N) as urea two weeks after plant emergence. No additional fertilizer was applied to the low N blocks. All trials under both low-N and high-N were under irrigation throughout the growing periods as required to avoid moisture stress. Field trials of the diversity panel were conducted by an augmented complete block design as described in the previous studies [17,20].

### 2.2. Measurements of Fatty Acid and Seed Oil Percentages

The phenotype data were collected from four different locations. For each site, seeds were harvested separately from four to six randomly selected plants in each line and used for measuring oil and fatty acid contents. The data from Bozeman were published in a previous report [17]. Using the same methods, relative abundance and composition of seed fatty acids were determined by analyzing fatty acyl methyl esters (FAMEs) using gas chromatography on a Shimadzu 2010 GC system (Shimadzu, Kyoto, Japan) with an Agilent HP-Innowax column (19091N–133; 30 m × 0.25 mm i.d. × 0.25 μm) (Agilent Technologies, Inc., Santa Clara, CA, USA). The oven temperature was programmed at 190 °C initially, followed by an increase of 20 °C/min to 250 °C and maintained for 3 min. Seed oil content was determined by a benchtop nuclear magnetic resonance seed analyzer (MQC23, Oxford Instruments, Concord, MA, USA) and presented as percentages of seed mass.

### 2.3. Marker Discovery and Validation

The whole genome resequencing data of the diversity panel were obtained previously [17]. The sequences were mapped to the recently assembled Suneson genome [18] using bowtie2 [21], and then the mapped reads were converted and filtered with SAMtools v1.14 [22] to include only those with a read depth of at least 10, an alignment score of 20 or more, and a base quality of 30 or more. BCFtools v1.16 [22] was then used to create mpileups and base calls for each alignment file. The resulting call files were combined, and the markers were filtered to keep only biallelic sites and remove genetic markers that had 20% or more missing data. This process was followed separately for SNPs and indels to produce two separate marker sets in the VCF format.

Linkage decay was estimated by using PLINK 1.9 [23] to calculate the genome-wide linkage statistics, which were plotted using R packages stringr [24], dplyr [25], and ggplot2 [26].

### 2.4. Population Structure

The circular maximum likelihood phylogenetic tree was created by combining the SNP and indel marker sets and importing them into R v4.4.1 [27]. Three essential R packages, “ape” [28], “vcfR” [29], and “phangorn” [30], were used to calculate and plot the tree. The subpopulation assignment was separated based on genetic distances illustrated in the phylogenetic tree.

The population structure was calculated using STRUCTURE v2.3 [31], and three separate structure runs were calculated. Run 1 used no population priors and assumed no admixture, run 2 used no population priors and assumed that there was admixture, and run three used population identity priors provided by the phylogenetic tree assignment and assumed that there was admixture. The STRUCTURE program was run inside an R v4.1 wrapper package “dartRverse” [32,33], which uses several sub-packages to run STRUCTURE and plot the results.

### 2.5. GWAS Program, Model Selection, and Settings

GWAS was performed in the GAPIT v3.5 R package [34]. To select the appropriate model for the study, a model pick test was conducted using seed oleic acid content from the Bozeman data set with several of the models available in the GAPIT package. The results of this initial test (Appendix A) revealed that the Fixed and Random Model Circulating Probability Unification (FarmCPU) [35] model or the Bayesian-information and Linkage-disequilibrium Iteratively Nested Keyway (BLINK) [36] model had enough sensitivity to drop false positive results and statistical power to report even rare variants. Of these two models, BLINK was chosen to proceed due to the increased statical power over FarmCPU, and BLINK avoided the assumption implicit in the FarmCPU model that causative variants are evenly spread over the genome.

The BLINK model accounts for population structure through setting a principal component analysis (PCA) count when the model is set up. After several rounds of testing, combined with measuring the percent variance explained per principal component, we settled on a PCA count of 4 to remove the most false positives while keeping more potential markers.

### 2.6. Pangenome Analysis

To take advantage of the available pangenome [37], a custom tool was built to use the list of associated markers as input, and output aligned genes from each member of the pangenome were classified into categories based on differences in nucleotide sequence and protein coding sequences. This tool is located on GitHub Enterprise Server 3.17.5 [38]. The tool functions by first parsing the supplied marker list into bed files with a user-supplied number of flanking bases. It then creates a FASTA file consisting of the mRNA sequences taken from the reference genome gene annotations that correspond to the bed file coordinates. The NCBI blast + tool [39] is then used to create a database for each pangenome member genome, and the reference mRNA FASTA files are compared against each pangenome member in turn using blastn searches. The best hit for each separate mRNA blast search is taken from each pangenome member, along with a user-supplied number of flanking sequence bases, and is placed into a new FASTA file. These files are aligned using MUSCLE ver. 5 [40] and classified into categories based on the number and type of differences (synonymous vs. nonsynonymous CDS differences, intron/non-coding only, no differences or no match) present between the aligned sequences. The resulting alignments are in FASTA format and can be viewed in any genome browser that supports MUSCLE alignments. The alignments can then be visually assessed to gauge whether the detected variations warrant further investigation.

## 3. Results

### 3.1. Variation of Seed Fatty Acid Composition in Different Growth Locations

Camelina seed oil characteristics varied widely across four growth locations (Figure 1). For example, linolenic acid (18:3) is the most abundant fatty acid in camelina seed. The average content was the highest at 36.3% in Bozeman, while it was lower in Sidney, Pullman, and Wuhan at 33.5%, 31.4%, and 23.2%, respectively (Figure 1). The population means were reflective of the relative environments of the locations where they were grown. During the typical growth season for camelina (April–July), Sidney is warmer (e.g., high 24.4 °C, low 9.6 °C in 2022) than either Bozeman (high 21.8 °C, low 5.7 °C) or Pullman (high 22.7 °C, low 7.9 °C). Sidney and Bozeman have comparable moisture levels, while Pullman has the lowest average monthly rainfall as indicated above (see Section 2.1). Wuhan has the warmest growth climate and highest moisture levels (high 27.9 °C, low 21.8 °C, 6.2 inches average monthly rainfall) (Appendix A). The most striking reduction of 18:3 with a concurrent increase of 18:2 was observed in Wuhan. Plants grown in Wuhan also accumulated high levels of saturated fatty acids 16:0 and 18:0. Stearic acid (18:0) was increased from 1–2% in other locations to almost 6% on average (Appendix A). Figure 2 further illustrates the correlations between environments (location, nitrogen treatment) and seed oil traits (oil content and fatty acid composition). Growth location (*p*-value < 2 × 10^−16^, F value 3730.4) and N treatments (*p*-value 7.78 × 10^−6^, F value 217.7) significantly affected seed oil content. A slight decrease in oil content averages across the population was observed in high-N fields at Pullman (high-N vs. low-N: 35.7%, 39.0%) but not Sidney (high-N vs. low-N: 30.7%, 30.5%). Environments, especially locations, also affected fatty acid composition, notably 18:0, 18:2, and VLCFAs. Of interest was the weak correlation between increased oil content and C-18 fatty acids and a strong negative correlation with VLCFAs. The other strong patterns were found among fatty acids, most strikingly the positive correlation between 16:0, 18:0, and 18:2, and the strong negative correlation each had with increased 18:3.

### 3.2. Marker Discovery and Population Structure Analysis

To evaluate the potential of the diversity panel for mapping QTLs associated with seed oil and fatty acids by GWAS, we analyzed the population structure based on molecular marker distributions. Using the recently assembled Suneson genome and the available resequencing data of the population [17], we discovered 203,319 SNP and 99,067 indel markers. These markers were then used to define the population structure, which could help eliminate false positives not related to the trait of interest. The population structure of the camelina diversity panel was previously studied, where the 212 accessions were grouped into four distinct sub-populations based on fewer markers [17]. Using the new combined SNP and indel marker sets, six or eight possible sub-populations were suggested, with either no admixture or with admixture. Taking into account a phylogenetic tree calculated with maximum likelihood as population identity priors (Appendix A), strong evidence supported seven populations based on ΔK (Appendix A). The results plotted using color to represent the percentage of subpopulation identity (Figure 3) reveal the level of admixture across the diversity panel.

To evaluate whether genomic diversity determines the fatty acid concentrations in the population across growth environments, we performed ANOVA using the location/trait combinations and the subpopulation identities, and the q matrix generated from the average population identity values computed by STRUCTURE. The results showed associations of fatty acid composition among seven subpopulations (Figure 4). Lines closely related to group 1 had significant differences in 18-carbon fatty acids, and other differences between genotypes could account for the performance of the diversity panel in the four diverse growing conditions. Principal component analysis (PCA) showed that the first five components only accounted for 17.5% of the total population diversity. The distribution of the population by components 1–3 revealed only a few distinct groups with small numbers of individuals inside (Figure 5), which either disappeared or changed membership by components 3–5. In each case, grouping based on the principal components was revealed, suggesting the presence of some strong identities. However, outside of the three main groups in plots A and B, and three different groups in plots C and D, the main bulk of the population is highly admixed (Figure 5). The association and PCA results indicated that genetic factors play a major role in the variation of fatty acid composition despite the environmental influence.

Linkage decay (LD) can be used as an indicator of the genomic range that a QTL is most highly associated with. The most common range definition is an LD below 0.1 or when at least half of the measured LD has decayed. Using the combined SNP and indel marker sets, we calculated the LD to be r^2^ = 0.25 at <10 kb, <0.10 at 50 kb, and leveled out at less than 0.05 between 100 and 200 kb (Figure 6). Based on these thresholds, a standard area of linkage for each marker of 20, 100, or 200 kb can be considered that houses candidate genes. Given ~1 gene per 4700 bp (approximate assembled genome size of 650 Mbp and 138 k gene models), regions of those sizes contain 4–5, 20–25, or 40–50 genes, respectively. Local linkage calculations performed by the GAPIT program showed much smaller linked regions, typically 6–8 kb for the SNP marker set and 2–4 kb for the indel marker set (Appendix A).

### 3.3. GWAS Identified Multiple QTLs for Camelina Seed Oil and Fatty Acid Contents

In the previous study, we identified only a limited number of significant markers associated with oil content and oleic acid (18:1) [17]. Performing GWAS using the new marker set and the additional phenotype data from the three new locations, we were able to reveal a total of 118 markers for oil content and major fatty acids in camelina seed across 31 trait/location combinations (Table 1). Eleven of the markers were associated with more than one trait/location, and one marker, SNP 1392, was found in the same genomic location as the previous GWAS results [17]. Our analysis also revealed substantial differences in marker associations between locations and nitrogen treatments (Figure 7). Significant overlap between growth locations was detected for several markers associated with 18:1. Markers for oil content and other fatty acids were identified in all four environments, though not for every N-treatment condition (Table 1). These results agreed with the observation that locations had major impacts on those traits, while they had little effect on 18:1 (Figure 2).

#### 3.3.1. QTLs for Oil Content and Linolenic Acid

Growing the camelina population in multiple environments resulted in the identification of markers associated with oilseed traits that were not revealed in the previous study conducted at one location [17]. The oil content QTLs were previously localized on chromosomes 2, 6, and 16, with the one on chromosome 16 being the most significant [17]. The current GWAS, however, detected different QTLs on chromosomes 7 and 14 (Table 2). More QTLs were discovered in other growth locations (Table 1), though not every nitrogen treatment, reflecting the significant effects of location on oil content (Figure 2).

This study also detected markers that were associated with 18:3, the most abundant fatty acid in camelina seed. These five markers were located on chromosomes 8, 14, and 19 (Table 2). Consistent with the 18:3 content influenced by locations, these markers were only found in one growth location (Sidney, MT, USA) and only when tested using either the high nitrogen or the averages of the low and high nitrogen treatment data. While the effect of one marker was low, the total effect was additive based on the number of markers/alleles an accession had that differed from the reference genome.

#### 3.3.2. Markers Associated with Oleic Acid Content

Oleic acid was least affected by environments (Figure 2), suggesting its strong heritability. Consistent with this observation, many QTLs for 18:1 were repeatedly detected across multiple environments. Those markers mostly resided on Chromosome 1. Of the 33 unique markers, eight were detected in multiple environments, most notably SNP 1392 (Table 3). GWAS using high-density markers in this study also revealed SNP 653, an additional marker on Chromosome 1 for increased oleic acid in the Sidney high-N trait/location combination (Figure 8). Besides, the Indel 458 marker was detected at both Sidney and Pullman sites. The markers presented in Table 3 have four distinct regions. When considering the mid-range interval of 50 kb for linked genomic regions, SNPs 591 and 653 have overlapping intervals, indels 458_1, 480, 483, and SNPs 739, 742, 753, 754, 763 all have overlapping intervals, and SNPs 1392 and 1558 have separate intervals. In addition, markers were detected associated with 18:2, whose content was closely correlated with 18:1, including SNP 591, SNP 1558, and indel 480 (Table 3).

We analyzed markers that were persistently associated with 18:1 across locations and N fertilization conditions (Table 3). To identify those markers, initially, the SNP and indel markers were not filtered to remove markers in high linkage disequilibrium with one another, nor were they binned. We used the GWAS model BLINK that accounts for linkage disequilibrium and reports the single most significant of the linked markers. At the beginning of the study, only the data from Bozeman were available, and GWAS with that data revealed only SNP1392. As additional field data became available, SNP 1392 was also found for four other trait/locations, and another marker SNP 653 was discovered using the Sidney high-N data. To better define a region where candidate genes could be found, we iterated runs of BLINK to find the most significant markers. Each time BLINK reported a significant marker in the region defined by SNPs 1392 and 653, that marker was removed from the data set and BLINK was run again. After only three runs, no additional markers could be found. This revealed SNP 753, a new marker for increased oleic acid, and SNP 653 again, but in the Sidney low-N treatment. When removing both SNPs 653 and 753, SNPs 754, 739, and 742 were discovered. When all these SNPs were removed, no additional SNPs were found on chromosome 1 associated with seed oleic acid. The same approach identified indels 483 and 458–1 (Appendix A). The accessions with at least seven of the markers, grouped into haplotype 2, typically had higher oleic acid concentrations across all trait/location/N-treatment combinations (Figure 8). These markers were therefore validated, as they linked to each other and to the 18:1 content.

### 3.4. Variation of Putative Candidate Genes Within Fatty Acid QTLs Among Pangenomes

The pangenome of *Camelina sativa* has recently been constructed, which contains 12 accessions belonging to different subpopulations. Their genomes were assembled at high quality using the latest technologies [37]. The pangenome therefore provides an additional tool to validate markers discovered above. When a putative marker and its flanking sequences are compared across the pangenome, markers that are present in pangenome members can be considered of high confidence. For example, one pangenome member, *Camelina sativa* var Svalof, corresponding to accession 35 in this study, has the G > A SNP 1392, as well as SNPs 653, 739, 742, 753, 754, and indel 458_1. These markers were therefore validated, making them part of the haplotype with increased seed oleic acid content.

To search for candidate genes that affect camelina seed fatty acid composition and oil content, we focused on the genomic intervals of 50 kb centered on each marker based on the linkage decay calculations (Figure 6). Markers that overlap when using this size of interval were considered as composite regions to prevent repeated analysis of the same gene. Within 118 QTLs, there are 2676 candidate genes. To narrow down the number of potential candidates, we compared coding sequences in marker regions between the Suneson and the pangenome to search for genes with similar inheritance patterns. We also used this comparison to prioritize gene candidates based on the type of differences that are linked to the marker. Results of this analysis are listed in Table 4, including 880 genes that have modifications to the deduced protein sequences in four or more of the pangenome members and 507 genes that have protein modifications in one to three pangenome members. Genes were classified in this way to match the observed inheritance pattern of the pangenome. Those genes having only silent modifications in the coding sequences (CDS) or modifications in non-coding regions (UTR and intron) are also listed, along with genes having either no matches among the pangenome or no differences between the pangenome and the reference Suneson genome. The classifications are conditionally exclusive, from more differences between pangenome members to fewer differences, such that genes with silent CDS differences may have differences in the non-coding regions as well. With these comparisons, genes with alterations in the protein-coding regions that lead to changes in the amino acid sequence of the CDS can be examined preferentially before those with only silent mutations.

## 4. Discussion

### 4.1. Successful QTL Mapping Depends on Both Genetic Markers and Trait Measurements

Natural variation provides valuable resources for understanding genetic mechanisms controlling important agronomic traits and integrating beneficial alleles into modern varieties through marker-assisted breeding and targeted genetic engineering [41]. Analyzing natural populations by GWAS is a powerful approach to understanding the genetic mechanisms underlying complex traits. In oilseed crops such as *Brassica napus*, this approach using a large inbred population identified comprehensive gene modules and predicted hundreds of genes significantly associated with seed oil content [42]. To effectively map QTLs, high marker quality is of paramount importance to trust the GWAS results, while a large marker quantity is required to improve the resolution of the interested chromosomal regions. Previous work mapping QTLs for camelina oilseed traits relied on a double haploid genome, DH55 [17]. Genetic markers found in the current study were derived from reads mapped to a higher-quality genome of a natural variety (Suneson) using improved genomics tools. This resulted in both increased marker quality and density, which ultimately revealed more markers associated with camelina oil traits that were not detected before, such as oil content and several fatty acids, and refined the regions that were previously identified, e.g., 18:1.

Another important aspect of QTL mapping is extensive phenotyping so that the genetic potential for the traits being studied can be fully revealed. The previous study indicated that camelina seed fatty acids displayed variation in the diversity panel; however, the range was small. Therefore, it was not surprising that fewer QTLs were detected and included only one fatty acid, 18:1 [17]. In this study, plants were grown in three additional geographically distinct locations. The data revealed dramatic effects of the growth environments on fatty acid composition, but also total seed oil content (Figure 1). Consequently, many QTLs were detected for these traits. Traits with low numbers of markers suggest that more of the variation was explained by the environment rather than genetic differences (Table 2). The phenotypic variation manifested by growth conditions under different locations and fertilization levels reflected both the genetics and the interactions between genotypes and environments. It has been reported that many of the yield-related traits in camelina were heavily influenced by environmental factors [43,44,45], and the seed fatty acid composition was especially susceptible to higher temperatures during seed filling [46]. Dissecting genetic mechanisms interacting with environments that determine these traits will require research with controlled conditions that separate different environmental factors, such as temperature and fertilization levels.

Fatty acid desaturation was the most significantly affected by environments (Figure 2), similar to reports in related species [47], reflecting the drastically different environments between locations, particularly temperatures. For example, Wuhan is the warmest and most humid of the four locations studied. Seeds produced there had much lower 18:3 (23%) compared to the highest measured average of 18:3 (36%) in Bozeman but increased stearic acid (18:0) and palmitic acid (16:0). Consequently, new QTLs were discovered for 18:3 besides several fatty acids (Table 1). Interestingly, environments had little effect on oleic acid (18:1), with a narrow range of ~14–18% for the location averages and an overall range of ~11–23% for all accessions and all locations. The QTL mapping results agreed with this observation as markers associated with 18:1 were persistently detected at all locations (Table 3). The dramatic changes in 18:1-derived unsaturated fatty acids, e.g., 18:3, can be attributed to the temperature sensitivity of fatty acid desaturases; for example, FAD3 activity is significantly decreased at high temperatures due to accelerated protein degradation [48]. The QTLs associated with 18:3 in this study, which do not contain genes encoding fatty acid desaturases, may uncover additional novel mechanisms regulating fatty acid modification under high temperatures.

### 4.2. Population Structure of the Camelina Diversity Panel

Defining the number of subpopulations in a diversity panel is important in GWAS since it reduces false positives by accounting for the genetic variation that does not relate to the trait being tested. Variations that exist outside of those subpopulations are easier to spot, leading to better results. An important caveat in determining population structure is that it may be skewed if only one type of marker is used. Each type of marker has its own linkage patterns due to different mutation rates, such as in the close genetic relative *Arabidopsis thaliana* [49]. To compute an accurate population structure for the camelina diversity panel, we combined both marker sets of SNPs and indels. This resulted in seven subpopulations, revised from either four or 13 previously determined using 161,301 SNPs [17]. We also tested the association of subpopulations with fatty acids and oil content (Figure 4). The identity with a subpopulation that shows this kind of association supports the population structure. However, such an association is unlikely to result in discovering markers that are linked to the subpopulation identity in question, since that identity is used as a covariate in the GWAS test to account for and remove population structure.

The population structure for this population is weak, as illustrated in Figure 3 and Figure 5. The weakness presents a problem because a highly admixed population with low genetic diversity can lead to false positives when the population structure cannot be adequately accounted for [50]. To overcome this, we evaluated the population in multiple growing locations/conditions to draw out the phenotypic responses reflecting even small genetic changes. The haplotype described by the markers associated with 18:1 on chromosome 1 did not conform to any single subpopulation identity and instead appeared to be randomly distributed throughout the population. When comparing the accessions that are classified as haplotype 2, no percent identity pattern was visible (Appendix A), except that there was little to no identity with group 1. This distribution of markers is expected, as those markers with inheritance patterns that followed the population structure closely would have been removed from consideration by the BLINK model. BLINK considers each marker in relation to its neighbors via a kinship matrix and the covariates we supplied by setting the PCA count or including the q matrix result from STRUCTURE. The markers that made up this haplotype were not all found in the same growing location or condition either, further validating the need to grow this population in diverse locations and conditions to capture a more complete picture of the phenotypic diversity, both inter- and intra-accession. It is reasonable to consider that other markers may be part of haplotype groups that are yet to be identified.

### 4.3. Linkage Decay and Pangenome Analyses May Aid in Identification of Candidate Genes Within QTLs

QTL regions usually harbor many genes that may include candidates for the complex traits being studied. Linkage decay, usually defined as the number of bases from the marker that must be measured before the maximum r^2^ is halved [51], determines the genomic region associated with the marker and its close genomic neighbors, but also delimits the regions to search for candidate genes. Linkage can decay swiftly depending on species, the type of population, and the markers used to calculate it. There may be only a few thousand bases from the marker before the r^2^ threshold is reached, or measured decay may take place over tens or hundreds of thousands of bases [52,53]. We measured camelina linkage decay and found a genome-wide average (Figure 6), which suggested three different estimates, 10, 50, and 100 kb from each side of the marker, for genomic regions associated with each marker, depending on which interpretation of the appropriate level of linkage decay is required for the delineation of a linked region. Searching for candidate genes may begin by investigating those inside the smallest region, 10 kb to either side of the marker, and expanding it if candidate genes prove difficult to identify. However, if candidate genes are not found in the first few thousand bases, the region of interest increases dramatically when the next level of linkage is considered. In the case of QTL for 18:1 on chromosome 1 bracketed by SNP 1392 and SNP 653, it can be taken separately to form individual regions of interest based on the linkage decay, but they could also bracket a larger area of ~2.1 Mbp that could contain multiple candidate genes. Though one notable gene in this region is *FAD2–2*, already known for its role in the biosynthesis of linoleic acid from oleic acid [4], it was not considered a causal gene since the sequences did not show variation in the population. The availability of the pangenome may also help identify candidate genes (Table 4). By comparing sequences across the pangenome, variations can be confirmed for the marker-trait associations, further narrowing the ranges harboring candidate genes.

Although the high-density markers and pangenomes narrowed the QTL regions and shortened the list of candidates, the goal of isolating a single candidate gene responsible for a complex trait has remained difficult to reach. Field studies strongly supported oleic acid being a highly heritable trait, and linkage of markers forming the haplotypes indicated the region harboring the candidate genes. It is also clear that there are a multitude of genes that are responsible for controlling the 18:1 content, given the several areas of the genome implicated by the QTLs in this study. The 18:1 concentration is mostly controlled by FAD2, but the complete network of genes regulating this pathway is unknown. The genetic variations in the marker-associated genomic regions and haplotypes are the peripheral influencers of these core genes, such as *FAD2*, each making up a small part of the regulatory system without containing the core gene inside the linked region. Other complementary approaches, such as gene expression studies between near-isogenic lines and targeted gene editing or misexpression, may facilitate isolation of the causal genes. The advancement of genomic studies in camelina, including improved annotation of the camelina genome for many genes of unknown functions, will enhance our ability to pinpoint the causal genes. The pangenome resource will also be improved over time as new genomes are added. Alignment classification tools such as the ones described in this study will be valuable for pinpointing genes that are the best potential candidates.

## 5. Conclusions

The development of high-quality molecular markers and repeated evaluation of a diverse population in different growth environments allowed for the identification of QTLs that are associated with camelina seed oil and major fatty acid contents. Field studies also revealed strong influences of environments on seed oil content and some fatty acids, highlighting the need to understand genetic mechanisms of seed oil biosynthesis for the development of camelina varieties adaptable to different environments. Although increasing the marker density improved the characterization of the natural population and narrowed the QTL regions, attempting to isolate single causal genes remained a challenge. Nevertheless, this study provides valuable materials and resources, such as molecular markers and plant lines for future studies to uncover new genetic mechanisms, which will explain variations of oil content and fatty acid composition in natural camelina populations and advise precision breeding strategies to improve oilseed traits.

## Figures and Tables

**Figure 1 biology-14-01199-f001:**
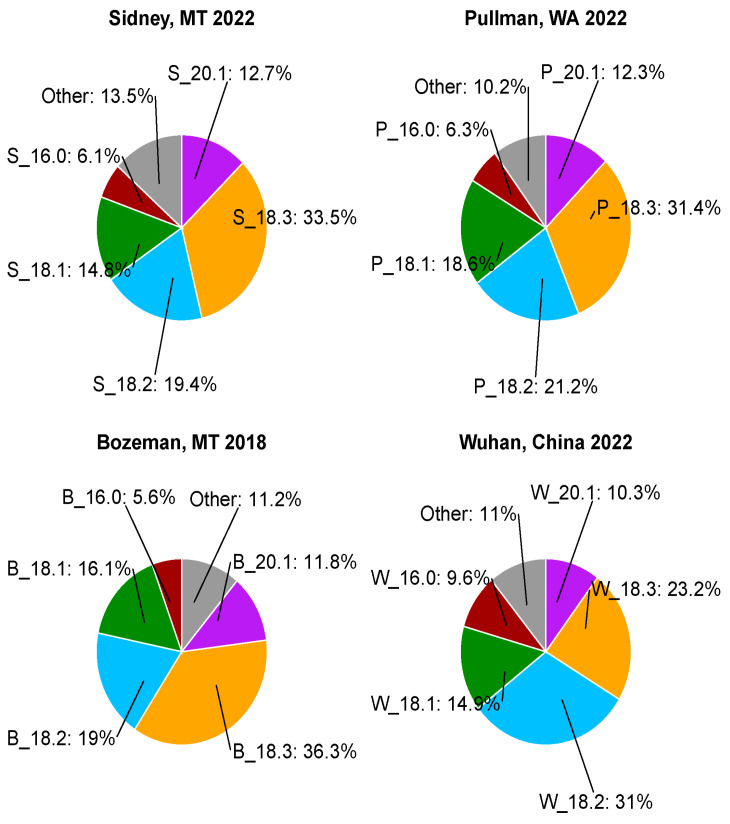
A summary of fatty acid composition means of the camelina population at each growth location. The data for Sidney and Pullman are the mean of the two nitrogen level treatments.

**Figure 2 biology-14-01199-f002:**
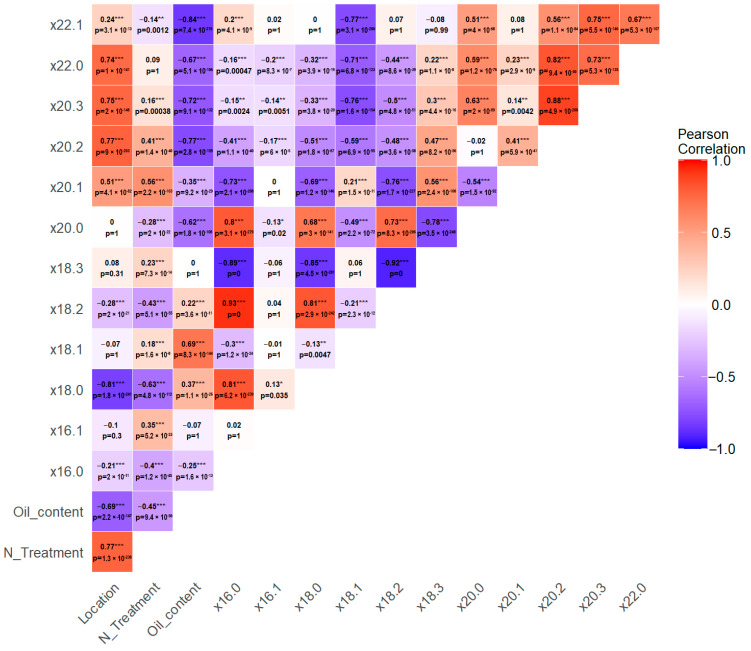
Pearson correlations of the location, nitrogen treatment, and oil and fatty acid contents. *p*-values have been subjected to a Bonferroni correction. Correlation *p*-values: * for < 0.05, ** for < 0.01, *** for < 0.001.

**Figure 3 biology-14-01199-f003:**
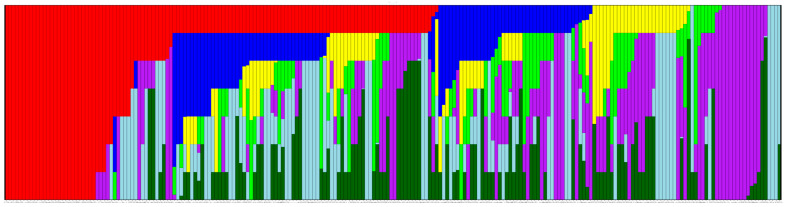
The population structure of the diversity panel, plotted with k = 7, suggested by STRUCTURE. The numeric identities are displayed as colors and grouped to show population identities of the diversity panel, illustrated from left to right (red, blue, yellow, green, purple, light blue, dark green).

**Figure 4 biology-14-01199-f004:**
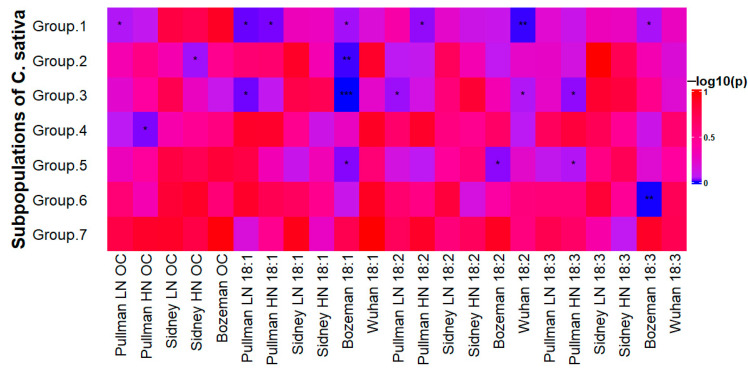
The heatmap of ANOVA results for seed fatty acids and oil content. The heatmap does not show an increase or decrease in fatty acids or oil content; instead, it shows that identity with one or more subpopulations is consistent with either a positive or negative difference in those traits over those without that identity. Correlation *p*-values: * for <0.05, ** for <0.01, *** for <0.001.

**Figure 5 biology-14-01199-f005:**
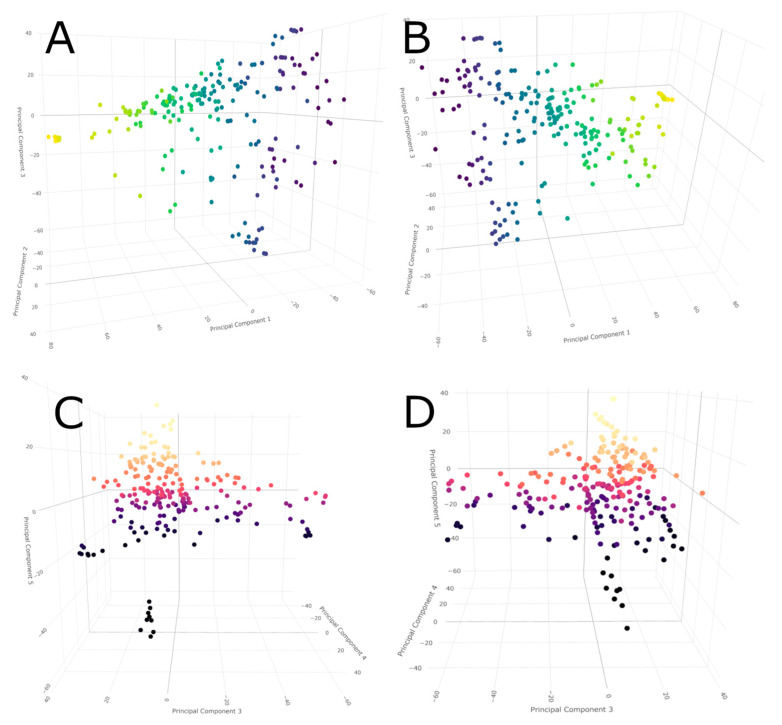
Principal component analysis showing 3D PCA plots of the diversity panel. (**A**,**B**) are of PCA components 1–3, and B is the view when the plot is rotated 180 degrees to give perspective on the other side of the plot. (**C**,**D**) are of PCA components 3–5, and D is rotated in the same manner as B. Coloration is a composite of the three selected PCA component values for each accession, combined to one scalar value and converted to RGB values.

**Figure 6 biology-14-01199-f006:**
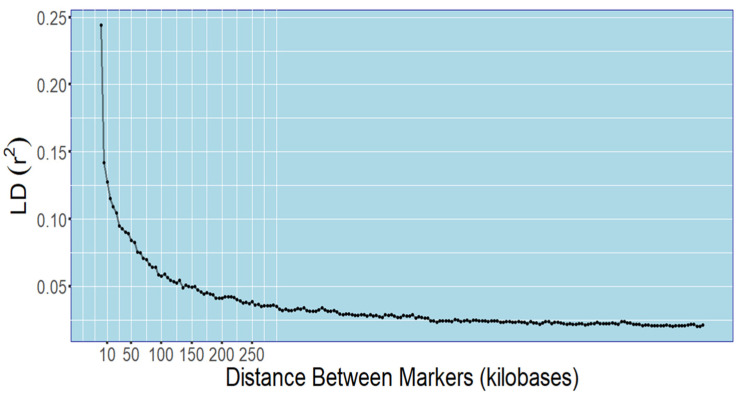
Linkage decay (LD) of combined SNP and indel markers, computed using Plink1.9 and plotted with R.

**Figure 7 biology-14-01199-f007:**
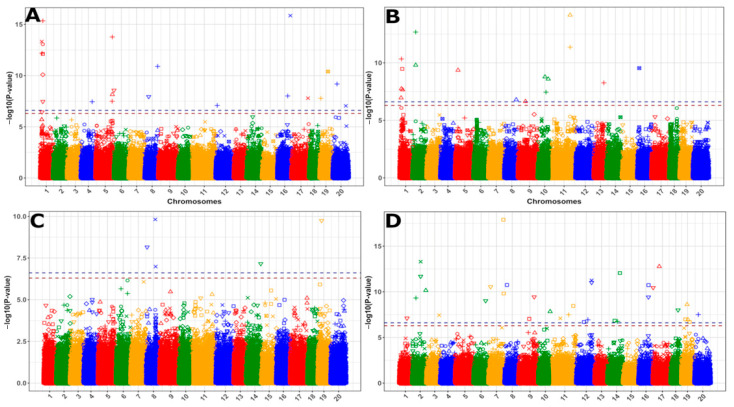
Manhattan plots of GWAS results for the diversity panel showing significant markers associated with eight trait/location/N-fertilization conditions. SNP and Indel results are plotted together. BZ for Bozeman, PM for Pullman, SD for Sidney, and WC for Wuhan, China. Nitrogen conditions: NA for nitrogen average, HN for added nitrogen, LN for no added nitrogen. (**A**–**C**) Markers associated with 18:1, 18:2, and 18:3, respectively. (**D**) Markers associated with seed oil content.

**Figure 8 biology-14-01199-f008:**
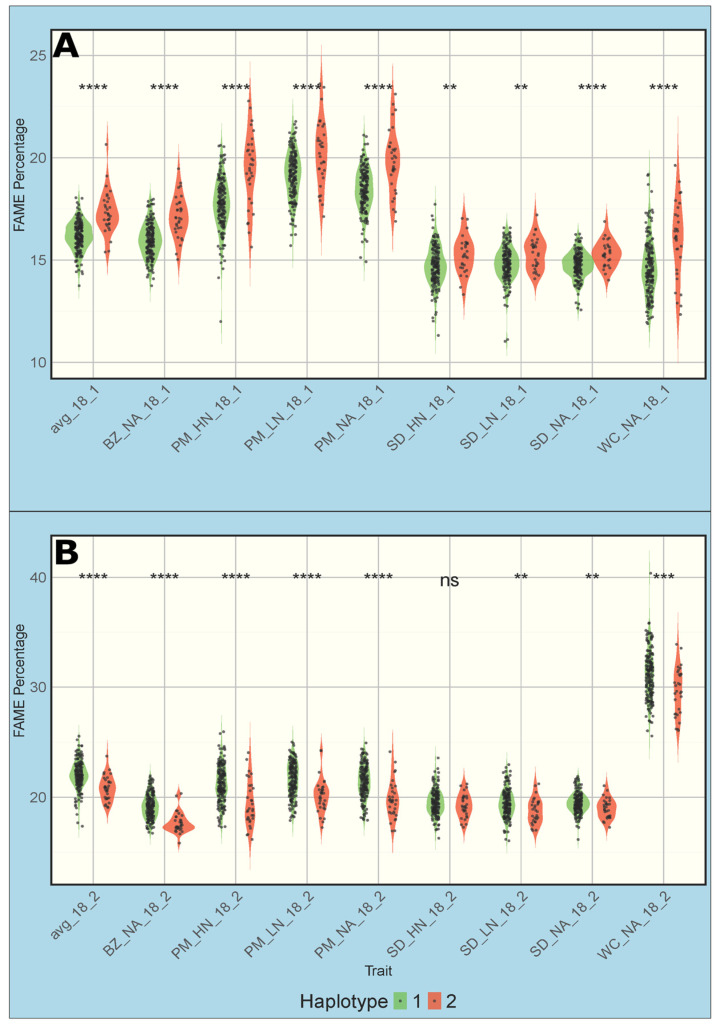
Comparison of 18:1 and 18:2 in seeds of two haplotypes. Haplotypes were defined as having at least seven of the markers associated with the change of these fatty acids in the region on chromosome 1 bounded by the SNP_591 and SNP_1392. (**A**) Violin plots of 18:1 contents across all trait/location/N-treatment combinations, separated by haplotype. (**B**) Same type of plots as A, except for 18:2. Significance calculated using one-way ANOVA. ** = *p*-value < 0.01, *** = *p*-value < 0.001, **** = *p*-value < 0.0001, ns = not significant.

**Table 1 biology-14-01199-t001:** Distribution of QTLs for fatty acids and oil content. The first number is of unique markers, and the second is of markers that are repeated for one or more locations/treatments.

Palmitic (16:0)	Stearic (18:0)	Oleic (18:1)	Linoleic (18:2)	Linolenic (18:3)	VLCFA (20–22:x)	Oil Content
10/1	16/1	33/8	17/0	5/0	37/1	34/0

**Table 2 biology-14-01199-t002:** Markers associated with seed oil content and linolenic acid (18:3).

Trait	Location, Nitrogen	Marker	Chr	Pos	*p*-Value	MAF	Effect	PVE (%)
Oil content *	SD, NA SD, HN	SNP 12699	2	17472554	2.04 × 10^−12^ 5.03 × 10^−14^	0.064 0.064	−1.175 −1.508	12.83 11.02
	SD, NA SD, HN	SNP 129397	12	31847058	9.75 × 10^−12^ 5.62 × 10^−12^	0.076 0.074	1.013 1.27	38.64 64.24
	BZ, NA	SNP 71383	7	30829468	1.25 × 10^−18^	0.162	−0.656	30.79
	BZ, NA	INDEL 74281	14	26078537	8.87 × 10^−13^	0.052	−0.736	40.03
	PM, NA	SNP 10199	2	7566564	4.84 × 10^−10^	0.074	−1.429	19.19
	PM, NA	SNP 196088	20	7751829	3.17 × 10^−8^	0.107	−1.127	11.48
	PM, LN	SNP 169383	17	13201950	1.70 × 10^−13^	0.2	−1.11	15.46
	PM, LN	SNP 15542	2	28383138	7.13 × 10^−11^	0.126	−1.38	15.03
18:3	SD, NA	SNP 72835	8	1844855	6.96 × 10^−9^	0.130	0.876	14.08
	SD, NA	SNP 151966	14	31754793	6.95 × 10^−8^	0.054	−1.165	24.04
	SD, NA	SNP 187379	19	7626530	1.78 × 10^−10^	0.127	−1.035	21.85
	SD, HN	SNP 78439	8	19496871	1.52 × 10^−10^	0.099	1.320	24.34
	SD, HN	SNP 79026	8	20320858	1.03 × 10^−7^	0.335	−0.703	18.84

* List of 10 most significant of the 34 unique markers. MAF (Minor Allele Frequency) shows how rare the allele is in the population, giving insight into how the variation among candidate genes may be distributed. Effect is the difference in average values vs. the population mean. The total effect is calculated by adding the effect shown per instance of the non-reference allele at each locus. PVE (percent variance explained) is how much of the variation of the trait is attributable to the presence of the marker.

**Table 3 biology-14-01199-t003:** Markers associated with oleic and linoleic acid on chromosome 1.

Marker	Pos	*p*-Value	MAF	Effect	PVE (%)	Trait, Location, N Level
SNP 591	2611818	1.11 × 10^−7^	0.110	−0.930	9.90	18:2, PM, LN
SNP 653	2712622	9.35 × 10^−11^ 4.70 × 10^−14^	0.0896 0.0911	1.027 0.980	57.77 34.41	18:1, SD, LN * 18:1, SD, HN
INDEL 458_1	3063926	4.45 × 10^−15^ 1.46 × 10^−17^ 1.83 × 10^−11^	0.117 0.118 0.114	1.318 0.869 0.688	35.18 21.45 19.81	18:1, PM, HN 18:1, SD, HN 18:1, SD, LN
SNP 739 **	3125731	1.90 × 10^−11^	0.101	0.852	51.55	18:1, SD, HN
SNP 742 **	3148736	3.14 × 10^−10^	0.075	1.197	52.80	18:1, SD, LN
INDEL 480	3198922	1.80 × 10^−8^	0.150	0.808	3.83	18:2, PM, LN
SNP 753 *	3225897	5.53 × 10^−10^	0.126	0.596	23.75	18:1, BZ, NA
SNP 754 **	3226206	4.06× 10^−15^	0.133	0.620	25.65	18:1, BZ, NA
INDEL 483	3230709	6.45 × 10^−13^	0.088	1.131	10.53	18:1, PM, NA
SNP 763	3247468	4.59 × 10^−11^	0.076	−1.673	29.76	18:2, PM, NA
SNP 1392	5000540	3.38 × 10^−10^ 3.41 × 10^−8^ 7.64 × 10^−13^ 4.49 × 10^−16^ 8.68 × 10^−14^ 8.21× 10^−11^	0.0744 0.0760 0.0744 0.0762 0.0762 0.0743	−1.589 0.682 0.968 1.738 1.839 1.069	60.53 50.53 61.13 65.50 29.93 59.36	18:2, BZ, NA 18:1, SD, NA 18:1, BZ, NA 18:1, PM, NA 18:1, PM, HN 18:1, SD, LN
SNP 1558	5899218	2.22 × 10^−8^	0.138	−0.934	14.65	18:2, PM, LN

* = revealed after SNP 1392 is removed. ** = revealed after SNP 1392 and SNP 753 are removed. The data is broken down by trait, then location: PM for Pullman, WA, SD for Sidney, MT, and BZ for Bozeman, MT. The nitrogen condition is LN for low, HN for high, and NA for average.

**Table 4 biology-14-01199-t004:** The variation of genes within QTLs in the pangenome. Genes were selected from the 50 kbp flanking sequence for each marker.

Associated Marker Set Tested	Total Genes	Four or More Pangenome Members with Protein Differences	One to Three Pangenome Members with Protein Differences	Silent CDS Differences	Differences in Non-Coding Regions Only	No Match or No Differences
All 118 fatty acid markers	2676	880	507	144	806	339
Markers from Table 3	254	56	61	24	101	12
SNP 1392	33	4	3	5	20	1

## Data Availability

All data are included in this publication.

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
