# Peer review of "Genomic Analysis of the Natural Variation of Fatty Acid Composition in Seed Oils of Camelina sativa"

_biology, 2025, doi:10.3390/biology14091199_

Round 1

Reviewer 1 Report

Comments and Suggestions for Authors

The manuscript by Decker et al. presents a GWAS analysis of 212 camelina accessions, focusing on oil and fatty acid composition and their interactions with environmental factors. It provides a comprehensive assessment of camelina oil profiles across four growth locations and successfully identifies hundreds of associated markers. The manuscript is well written, and I have the following suggestions to further strengthen the discussion.

Introduction: 2nd paragraph Lines 15-18: “only achieved by inserting the required pathway into camelina, suggesting the contribution of endogenous seed lipid mechanisms” Elaborate a bit, it is confusing.

Figure 5: cannot read, legends/labels are too smalls, the color code for different groups is not clear. It may be helpful to remove the label of camelina accessions from the figure and focusing on the clusters and provide important accession information in a supplementary file.

Discussion 4.1:  2nd paragraph, lines 4–8 from the end: One possible explanation for the observed changes in fatty acid composition across locations with different temperatures is that FADs are temperature-sensitive. FADs are more active at low temperatures and less active at high temperatures. Such discussion can be included.

Discussion on Figure 2: Did the authors observe any association between oil content and nitrogen fertilization? Did they measure the protein content of the seeds? If so, are there any relationships among oil content, protein content, and nitrogen fertilization?

Did the authors observe any association between oil content and any fatty acid profile?

Discussion 4.3: Marker analysis and single gene identification: I understand the challenge of narrowing QTL regions and pinpointing a single causal gene. However, it would still be worthwhile to present 1–2 representative genes in greater detail. For example, FAD2 variations were observed. What specific changes in FAD2 are associated with high 18:1 content? Were these changes in the promoter, coding sequence, or other regions? Including a sequence alignment would help illustrate the cause and demonstrate the powerful of identified markers.

Minor comments: format issues/line breaks on pages 4, 5, 7, 8.

Author Response

Thank the reviewer for critical comments and very good suggestions. We appreciate their time and service for the scientific  field.  Our responses are below.

Comment 1: Introduction: 2nd paragraph Lines 15-18: “only achieved by inserting the required pathway into camelina, suggesting the contribution of endogenous seed lipid mechanisms” Elaborate a bit, it is confusing.

Response 1: This phrase is revised for clarity. It reads and highlighted in revision: Metabolic engineering of fish oil-type long chain PUFAs including eicosapentaenoic acid (EPA) and docosahexaenoic acid (DHA) was achieved in Arabidopsis by reconstructing the biosynthetic pathway comprising heterologous fatty acid desaturase and elongase genes [13]. The high levels of such LC-PUFAs that matched those found in marine fish oils were only achieved by expressing those genesin camelina, suggesting the contribution of endogenous seed lipid mechanisms in camelina for efficient accumulation of DHA and EPA [14].

Comment 2. Figure 5: cannot read, legends/labels are too smalls, the color code for different groups is not clear.

Response 2. Fig. 5 has been replaced with better coloration.

Comment 3. Discussion 4.1:  2nd paragraph, lines 4–8 from the end: One possible explanation for the observed changes in fatty acid composition across locations with different temperatures is that FADs are temperature-sensitive. FADs are more active at low temperatures and less active at high temperatures. Such discussion can be included.

Response 3. Thanks for the suggestion. The discussion for this possible explanation is included as highlighted: The dramatic changes in 18:1-derived unsaturated fatty acids, e.g. 18:3, can be attributed to the temperature sensitivity of fatty acid desaturases;for example, FAD3 activity is significantly decreased at high temperatures due to accelerated protein degradation [24]. The QTLs associated with 18:3 in this study, which do not contain genes encoding fatty acid desaturases, may uncover additional novel mechanisms regulating fatty acid modification under high temperatures.

Comment 4. Discussion on Figure 2: Did the authors observe any association between oil content and nitrogen fertilization? Did they measure the protein content of the seeds? If so, are there any relationships among oil content, protein content, and nitrogen fertilization?

Response 4. We did observe the correlation between oil content and N fertilization, which we added in the revision highlighted. Growth location (p-value < 2e-16, F value 3730.4) and N treatments (p-value 7.78e-6, F value 217.7) significantly affected seed oil content. A slight decrease in oil content averages across the population was observed in high-N fields at Pullman (high-N vs low-N: 35.7%, 39.0%), but not Sidney (high-N vs low-N: 30.7%, 30.5%).  We did not include data on seed protein (or N) content because this report focuses on oil characteristics. As reported in other oilseeds and our own data in previous work, seed oil content is negatively correlated with protein content. 

Comment 5. Did the authors observe any association between oil content and any fatty acid profile?

Response 5. Those correlations were included in Fig. 2, and in the text as stated: Of interest is the weak correlation between increased oil content and C-18 fatty acids and a strong negative correlation with VLCFAs.

Comment 6. Discussion 4.3: Marker analysis and single gene identification: I understand the challenge of narrowing QTL regions and pinpointing a single causal gene. However, it would still be worthwhile to present 1–2 representative genes in greater detail. For example, FAD2 variations were observed. What specific changes in FAD2 are associated with high 18:1 content? Were these changes in the promoter, coding sequence, or other regions? Including a sequence alignment would help illustrate the cause and demonstrate the powerful of identified markers.

Response 6. We agree with this suggest, and in fact we have been trying to pinpoint candidate genes but so far we cannot identify any one with confidence. In the revision, we clarified as: Though one notable gene in this region is FAD2-2, already known for its role in biosynthesis of linoleic from oleic acid [30], it was not considered a causal gene since the sequences did not show variation in the population.

Reviewer 2 Report

Comments and Suggestions for Authors

Dear Authors,

Your article work is very good. I do have some comments and suggestions to help further improve your work. All the comments and suggestions in the attached file.

Author Response

We appreciate the reviewer's time and suggestions for improving our manuscript. Our responses are below.

Comment 1. The title "Genomic Insights into the Variation of Fatty Acid Composition in Seed Oils of Camelina sativa" is scientifically sound and contextually appropriate, but it could be refined slightly for clarity. Here is my proposed title. Please feel free to adjust it if needed. Genomic Analysis of Fatty Acid Composition Variability in Camelina sativa Seed Oils. 

Response 1. Great suggestion. The title is changed to Genomic Analysis of the Natural Variation of Fatty Acid Composition in Seed Oils of Camelina sativa.

Comment 2. Keywords I suggest add parents and offspring, if possible. 

Response 2. We added two keywords "pangenoe" and "population structure". This research does not involve offspring.

Comment 3. Introduction Overall, the introduction was defined clearly. Add the camelina global production amount, according to the FAO Stat. (identification of only one QTL associated) Change to (identification of only one quantitative trait locus (QTL) associated with. Should be each abbreviation described at first present. 

Response 3. As camelina is an emerging crop, production data is scarce, therefore we did not add that type of data. The abbreviation QTL has been spelled out in the revision.

Comment 4. Why the amount of fertilizers was not applied in other two locations (Bozeman and Wuhan). What reason behind this. 

Response 4. Bozeman and Wuhan studies were independently conducted. New locations at Pullman and Sidney were treated with different N fertilization. Data were analyzed separately.

Comment 5. The quality of figures 3and 5 are too low, put high resolution figure if possible. 

Response 5. Figures have been updated.

Comment 6. Some of the references are too old, use UpToDate references, if available. Reference number 17, was used in a lot of time, why? 

Response 6. References have been updated, if new ones are available. It should be noted that key concepts were referred to older literature due to their originalities and significance. Ref 17 was used frequently because this study is the continuation of that one with significant technical and analytical improvements, and therefore more findings.

Reviewer 3 Report

Comments and Suggestions for Authors

Dear Authors,

Manuscript Title: Genomic Insights into the Variation of Fatty Acid Composition in Seed Oils of Camelina sativa

The manuscript explores genetic factors influencing fatty acid composition in Camelina sativa using genome-wide association studies (GWAS) across multiple environments. The work is timely and relevant, especially given the growing interest in camelina as a sustainable oilseed crop for biofuel and nutritional applications. The integration of high-density molecular markers with multi-location phenotyping is a clear strength. However, several areas require clarification, refinement, and deeper analysis to strengthen the manuscript and enhance its scientific contribution.

Comments

  1. Novelty and Significance
    • While the study identifies new QTLs and confirms previously reported loci, the novelty could be emphasized more clearly in the introduction and discussion. Specifically, the manuscript should articulate how these findings advance the field beyond earlier GWAS studies on camelina.
    • A clearer statement on how this work may inform breeding programs (e.g., for high-oleic, low-PUFA oils) would improve the applied impact.
  2. Methodological Clarity
    • The Materials and Methods section is comprehensive but at times too technical for reproducibility. For example, more details are needed regarding replication: How many biological replicates per accession per location were analyzed? Were field trials randomized or blocked to account for environmental variation?
    • The statistical approaches (FarmCPU vs. BLINK models) are mentioned, but the rationale for model choice and parameter tuning (e.g., number of principal components) should be justified with clearer references or supporting data.
    • The description of the pangenome analysis is very brief. More methodological detail is needed for reproducibility (e.g., how candidate gene prioritization was validated).
  3. Data Presentation and Interpretation
    • Figures 1–8 provide useful summaries, but many lack sufficient resolution and detail in the legends (e.g., units, explanation of abbreviations, criteria for significance).
    • The interpretation of environmental effects (e.g., the reduction of 18:3 in Wuhan) is interesting but could be expanded. Including a discussion of genotype × environment interactions would strengthen the argument.
    • The discussion sometimes extrapolates beyond the data—for instance, linking QTLs directly to candidate genes without functional validation. The authors should present such associations as putative rather than definitive.
  4. Limitations and Future Work
    • The study does not explicitly discuss its limitations. Examples include: (a) reliance on statistical associations without functional validation of candidate genes, (b) limited coverage of environmental stress conditions, and (c) challenges posed by high admixture in the diversity panel.
    • Future directions could be better highlighted, such as integrating GWAS results with transcriptomic or metabolomic datasets, or validating candidate loci via CRISPR or near-isogenic lines.

Minor Comments

  1. Figures and Supplementary Data
    • Ensure all supplementary figures (S1–S5) are referenced in the main text at appropriate points.
    • Manhattan plots (Fig. 7) should be improved for clarity—significance thresholds and marker density need clearer labeling.
  2. References
    • The reference list is strong but could include more recent GWAS/QTL studies in oilseeds beyond camelina to broaden context.
    • Formatting should be double-checked for consistency with journal style.

Recommendation

Major Revision
The manuscript presents valuable results with clear potential impact. However, improvements are needed in methodological transparency, figure clarity, discussion balance, and highlighting of novelty. Addressing these points will significantly strengthen the paper and make it suitable for publication.

Regards,

Comments on the Quality of English Language

    • The manuscript would benefit from careful language editing. Some sentences are long and complex, reducing readability.
    • Technical terms (e.g., “linkage decay,” “haplotypes”) should be briefly explained for a broader readership.

Author Response

We thank very much for the reviewer's time spending on this work and providing constructive suggestions to improve the manuscript. Our responses to the major comments are as below. Minor comments have been incorporated into the extensively revised manuscript.

Comment 1. Novelty and Significance. A clearer statement on how this work may inform breeding programs (e.g., for high-oleic, low-PUFA oils) would improve the applied impact.

Response 1. The novelty of this study has been stated in the revision. Particularly we have included some languages in the Introduction and Conclusion sections.

Comment 2 on methodologies. 

  1. The Materials and Methods section is comprehensive but at times too technical for reproducibility. For example, more details are needed regarding replication: How many biological replicates per accession per location were analyzed? Were field trials randomized or blocked to account for environmental variation?
  2. The statistical approaches (FarmCPU vs. BLINK models) are mentioned, but the rationale for model choice and parameter tuning (e.g., number of principal components) should be justified with clearer references or supporting data.
  3. The description of the pangenome analysis is very brief. More methodological detail is needed for reproducibility (e.g., how candidate gene prioritization was validated).

Response 2.

  1. The following have been added: Field trials of the diversity panel were conducted by an augmented complete block design as described in the previous studies [17,20]. ... The phenotype data were collected from four different locations. For each site, seeds were harvested separately from four to six randomly selected plants in each line and used for measuring oil and fatty acid contents.
  2. GWAS statistical methods have been re-written in section 2.5 with references.
  3. Pangenome methods have been added with much details in section 2.6. Results are also discussed in section 3.4.

Comment 3. Data Presentation and Interpretation

  1. Figures 1–8 provide useful summaries, but many lack sufficient resolution and detail in the legends (e.g., units, explanation of abbreviations, criteria for significance).
  2. The interpretation of environmental effects (e.g., the reduction of 18:3 in Wuhan) is interesting but could be expanded. Including a discussion of genotype × environment interactionswould strengthen the argument.
  3. The discussion sometimes extrapolates beyond the data—for instance, linking QTLs directly to candidate genes without functional validation. The authors should present such associations as putative rather than definitive.

Response 3. 

  1. Figures have been updated with clarity and resolution. 
  2. Discussions on 18:3 variability and explanations, and the genotype-environment interactions have been expanded in section 4.1.
  3. Our studies have mapped QTL regions but have not identified definitive candidate genes. Discussions have been extensively revised integrating this valuable comment.

Comment 4. Limitations and Future Work

  1. The study does not explicitly discuss its limitations. Examples include: (a) reliance on statistical associations without functional validation of candidate genes, (b) limited coverage of environmental stress conditions, and (c) challenges posed by high admixture in the diversity panel.
  2. Future directions could be better highlighted, such as integrating GWAS results with transcriptomic or metabolomic datasets, or validating candidate loci via CRISPR or near-isogenic lines.

Response 4. 

  1. The discussions have been extensively revised as commented, particular in the genotype-environment section 4.1 and 4.3 (highlighted).
  2. Future directions are stated in the revised section 5 conclusions. 

Reviewer 4 Report

Comments and Suggestions for Authors

Overall, the study is well-structured; however, certain aspects require further clarification, additional supporting information, or revisions to strengthen the manuscript.

1. The study does not explain how these findings will contribute to breeding strategies, functional gene discovery, or industrial applications.

2. In material and methods, it is recommended to add a detailed table having information on number of plants on each site.

3. The age of the plants and the soil condition with parameters need to be mentioned in a table.

4. In result section, there is a line "Growth environments (location and N treatment) significantly affected seed oil content", but the p-value and f-value are not mentioned in support of the "significantly affected" term.

5. The person correlations shows good correlations, but is this corrections are significant (p<0.05) ? 

6. In methods section the detailed GC conditions need to be added.

7. In result section, need to define the terms "warmer", "cooler", "dried".

8. The conclusion reads more like an extended results summary rather than a synthesized conclusion. The practical impact (e.g., potential breeding applications, industrial utility in bioenergy and nutrition) could be expanded to better connect findings to real-world outcomes. The last sentence briefly notes that the study provides materials for future research. This could be strengthened by explicitly suggesting directions, such as integrating multi-omics approaches, gene editing, or breeding strategies for oil improvement.

Author Response

We thank the reviewer for spending time and providing suggestions to improve the manuscript. Our responses are below:

Comment 1. The study does not explain how these findings will contribute to breeding strategies, functional gene discovery, or industrial applications.

Response 1. Reverent texts have been added in "Introduction" and "Conclusion". They are highlighted in the revised manuscript.

Comment 2. In material and methods, it is recommended to add a detailed table having information on number of plants on each site.

Response 2. We added some details of the studies including number of plants on each site, highlighted in section 2.2.

Comment 3. The age of the plants and the soil condition with parameters need to be mentioned in a table.

Response 3. Seeds were harvested at maturity. We did not measure soil properties, but general soil conditions have been mentioned in previous studies:

Barnes, E.M., C.T. Yin, D. Schlatter, H. Peng, C. Willmore, C. Lu, S.G. Tringe, and T.C. Paulitz. Legacy Effects of Cropping System and Precipitation Influence the Core Microbiome. Phytobiomes Journal, 2025. 9: 327-339.

Etesami, Maral and Gautam, Shreya and Franck, William and Franck, Sooyoung and Arshad, Tehmina and Lu, Chaofu and Chen, Chengci, Variation of Yield Performance and Nitrogen Use Efficiency in Camelina Genotypes Under Low and High Nitrogen Environments. http://dx.doi.org/10.2139/ssrn.4972242

King, K., H. Li, J. Kang, and C. Lu. Mapping quantitative trait loci for seed traits in Camelina sativa. Theoretical and Applied Genetics, 2019. 132: 2567-2577.

Comment 4. In result section, there is a line "Growth environments (location and N treatment) significantly affected seed oil content", but the p-value and f-value are not mentioned in support of the "significantly affected" term.

Response 4. They have been added in revision: Growth location (p-value < 2e-16, F value 3730.4) and N treatments (p-value 7.78e-6, F value 217.7) significantly affected seed oil content. A slight decrease in oil content averages across the population was observed in high-N fields at Pullman (high-N vs low-N: 35.7%, 39.0%), but not Sidney (high-N vs low-N: 30.7%, 30.5%).

Comment 5. The person correlations shows good correlations, but is this corrections are significant (p<0.05)? 

Response 5. Fig. 2 now includes significance.

Comment 6. In methods section the detailed GC conditions need to be added.

Response 6. Detals have been added: relative abundance and composition of seed fatty acids was determined by analyzing fatty acyl methyl esters (FAMEs) using gas chromatography on a Shimadzu 2010 GC system with an Agilent HP-Innowax column (19091N-133; 30 m x 0.25 mm i.d. x 0.25 mm). The oven temperature was programmed at 190°C initially followed by an increase of 20 °C/min to 250 °C and maintained for 3 min.

Comment 7. In result section, need to define the terms "warmer", "cooler", "dried".

Response 7. They are defined in the revision: During the typical growth season for camelina (Apr-Jul), Sidney is warmer (e.g., high 24.4°C, low 9.6°C in 2022) than either Bozeman (high 21.8°C, low 5.7°C) or Pullman (high 22.7°C, low 7.9°C). Sidney and Bozeman have comparable moisture levels, while Pullman has the lowest average monthly rainfall as indicated above (see Methods). Wuhan has the warmest growth climate and highest moisture levels (high 27.9°C, low21.8°C, 6.2 inches avg monthly rainfall). 

Comment 8. The conclusion reads more like an extended results summary rather than a synthesized conclusion. The practical impact (e.g., potential breeding applications, industrial utility in bioenergy and nutrition) could be expanded to better connect findings to real-world outcomes. The last sentence briefly notes that the study provides materials for future research. This could be strengthened by explicitly suggesting directions, such as integrating multi-omics approaches, gene editing, or breeding strategies for oil improvement.

Response 8. The whole section of Conclusion has been revised as suggested.